# New Cadinane Sesquiterpenes from the Stems of *Kadsura heteroclita*

**DOI:** 10.3390/molecules24091664

**Published:** 2019-04-28

**Authors:** Liang Cao, Nuzhat Shehla, Shumaila Tasneem, Mengru Cao, Wenbing Sheng, Yuqing Jian, Bin Li, Caiyun Peng, M. Iqbal Choudhary, Duan-fang Liao, Wei Wang

**Affiliations:** 1TCM and Ethnomedicine Innovation & Development International Laboratory, Innovative Materia Medica Research Institute, School of Pharmacy, Hunan University of Chinese Medicine, Changsha 410208, China; caoliang520945@126.com (L.C.); shehla_chm@hotmail.com (N.S.); tasneemshum@gmail.com (S.T.); caomengru0913@126.com (M.C.); wbs626@126.com (W.S.); cpujyq2010@163.com (Y.J.); libin_hucm@hotmail.com (B.L.); paudy@126.com (C.P.); dfliao@hnucm.edu.cn (D.-f.L.); 2H.E.J. Research Institute of Chemistry, International Center for Chemical and Biological Sciences, University of Karachi, Karachi 75270, Pakistan; aurahman786@gmail.com

**Keywords:** *Kadsura heteroclita*, Schisandraceae, sesquiterpenes, antioxidant, Tujia ethnomedicine

## Abstract

As part of our continual efforts to exploit ‘Tujia Ethnomedicine’ for their pharmacophoric functionalities, we herein investigated *Kadsura heteroclita* collected from a deep Wulin mountain area in northern Hunan province. The current study resulted in the isolation of three new sesquiterpenes: 6α,9α,15-trihydroxycadinan-4-en-3-one (**1**), (+)-3,11,12-trihydroxycalamenene (**2**), (–)-3,10,11,12-tetrahydroxy-calamenene (**3**), along with four known sesquiterpenes (**4**–**7**), and a cytochalasin H (**8**). Their chemical structures were elucidated by 1D-, and 2D-NMR spectroscopy, and HRESI-MS, CD spectrometry. The antioxidant, and cytotoxic activities of the compounds were evaluated. Compound **8** exhibited a strong antioxidant effect with an IC_50_ value of 3.67 µM on isolated human polymorphonuclear cells or neutrophils.

## 1. Introduction

In the last few decades, phytochemical work on the family Schisandraceae has resulted in a seminal contribution to novel structural frameworks for triterpenoidal skeletons [1,2,3,4]. Recently, some promising sesquiterpenes has also been reported from the plants of family Schisandraceae [5,6,7,8,9,10]. In our previous research, we reported the identification of several bioactive lignanoids from the genus *kadsura* [11] that highlight the immense diversity in the chemical constituents of these plants as well as in their important biological activities. Herein, we report the identification of three new sesquiterpenoids (**1**–**3**) from *kadsura heteroclita*, amongst compound **1** was a structural analogue of our previously reported compound **4,** while compounds **2**–**3** possess calamenene core skeletons, which are identified for the first time from this species.

With the major distribution in China, genus *Kadsura* (Schisandraceaea) is represented by sixteen species, half of which are mainly distributed in the southwest and southeast part of China [12]. *Kadsura heteroclita* is a vine plant primarily distributed in the southwest part of China, has been used in Traditional Chinese Medicine (TCM) as “Hai Feng Teng” or “Ji Xue Teng” for the treatment of menstrual irregularities and blood deficiencies [13], by activating the blood circulation, relieving pain, and eliminating dampness. In the “Hu Ping” and “Xue Feng” mountain areas of northwest Hunan province, the stems of *K. heteroclita* are used to treat traumatic injuries and rheumatoid arthritis by one of the Chinese ethnic minorities known as the Tujia people [14,15]. Traditionally they named the stems of *K. heteroclita* as “Xue Tong” which means “circulating the blood system”, which is used to treat blood related diseases [16]. These features triggered our research enthusiasm to explore the bioactive metabolites from this species that can be related to its clinical application. The previous work implied that the plant contains triterpenoids [17,18,19], lignans [20,21,22] and sesquiterpenes [6]. In the present work, we reported three new sesquiterpenes (**1**–**3**) (Figure 1) along with four known sesquiterpenes 6α,9α-dihydroxycadinan-4-en-3-one (**4**) [6], isoepicubenol (**5**) [23], cryptomeridiol (**6**) [24], 1β,4β,7α-trihydroxyeudesmane (**7**) [25,26], and a cytochalasin H (**8**) [27,28] isolated from the stems and roots of *K. heteroclita* as shown in Figure 1.

## 2. Results and Discussion

### 2.1. Structure Characterization of the Isolated Compounds *(**1**–**3**)*

6α,9α,15-Trihydroxycadinan-4-en-3-one (**1**) (Figure 2) was isolated as a white amorphous powder. The molecular formula was established as C_15_H_22_O_4_ from its HRESI-MS (*m/z* 289.1419, [M + Na]^+^ calcd. 289.1410), indicating five degrees of unsaturation. The ^1^H-NMR spectrum (Table 1) the revealed presence of one secondary group (δ_H_ 1.00, d, *J* = 6.9), a tertiary methyl group (δ_H_ 1.83, s), one olefinic methine group (δ_H_ 6.85, s), and a terminal olefinic methylene group (δ_H_ 4.77, 4.90, brs each) in the structure. The ^13^C-NMR and DEPT NMR spectra of compound **1** indicated fifteen carbon signals attributed to one carbonyl carbon (δ_C_ 200.5), four olefinic carbons (δ_C_ 155.2, 146.6, 137.0, 114.6), and three oxygenated carbons (δ_C_ 73.3, 71.9, 59.5). Apart from these signals, three methines (δ_C_ 37.3, 48.6, 50.2), two methylenes (δ_C_ 36.8, 37.0), and two methyl groups (δ_C_ 15.0, 23.0) were also observed. Besides the two double bonds and one carbonyl, the remaining two unsaturation degrees together with the chemical shift patterns suggested that the compound is a bicycle sesquiterpene based on a cadinane skeleton. A downfield chemical shift of an olefinic proton (δ_H_ 6.85, s) conjugated with carbonyl carbon (δ_C_ 200.5) showed the presence of an α-β unsaturated ketone moiety in the skeleton of **1**.

By carefully comparing the ^1^H- and ^13^C-NMR chemical shifts data (Table 1) of **1** with that of known compound 6*α*,9*α*-dihydroxycadinan-4-en-3-one (**4**), which was also reported previously from *K. heteroclita* [6], we revealed the major difference between the two structures. The chemical shift of C-15 (δ_C_ 15.3, δ_3H_ 1.81, 3H) corresponding to a methyl group in **4** changed to oxygenated methylene (δ_C_ 59.5, δ_2H_ 4.16, 2H) in **1**. This deduction was supported by HMBC correlations of H_2_-15 with C-3 (δ_C_ 200.5), C-4 (δ_C_ 137.0), and C-5 (δ_C_ 155.2). An isopropenyl unit was also identified in the skeleton on the basis of the presence of terminal methylene (δ_C_ 114.6, δ_H_ 4.77, 4.90 brs each), a tertiary methyl signal (δ_C_ 23.0, δ_3H_ 1.83, 3H) along with olefinic quaternary carbon (δ_C_ 146.6). The HMBC correlations of H_3_-13 (δ_H_ 1.83) to C-7 (δ_C_ 48.6), C-11 (δ_C_ 146.6), and C-12 (δ_C_ 114.6), and of H_2_-12 (δ_H_ 4.77, 4.90 brs each) with C-7 (δ_C_ 48.6) and C-13 (δ_C_ 23.0), and HMBC correlation of H-7 (δ_H_ 2.54) with C-13 helped us to assign the isopropenyl unit at the C-7 position. The HMBC correlation of H_3_-14 (δ_H_ 1.00) to C-1 (δ_C_ 50.2), C-9 (δ_C_ 71.9), and C-10 (δ_C_ 37.3), along with COSY cross peaks between H_3_-14 and H-10 (δ_H_ 2.14) revealed a hydroxyl group attached to C-9. The HMBC signals of H_2_-2 (δ_H_ 2.39, 2.46 dd each) and H_2_-8 (δ_H_ 2.00, 1.65, m each) with C-6 (δ_C_ 73.3) supported the presence of another hydroxyl group located at C-6, thus the planar structure of compound **1** has been determined (Figure 2).

The relative configuration of **1** was determined by NOESY spectra in a combination of coupling constant analysis. The observed NOESY correlations between the spin systems H_3_-14 (δ_H_ 1.00)/H-9 (δ_H_ 3.47), H-9/H_β_-8 (δ_H_ 1.65), H_β_-8/H-7 (δ_H_ 2.54), and H-7/H-9 suggested that H-9, H-7, CH_3_-14, and H_β_-8 were positioned in β-orientation, while correlations of H-1 (δ_H_ 2.21)/H-10 (δ_H_ 2.14), H-10/H_α_-8 (δ_H_ 2.00), H_α_-8/H_3_-13 revealed that H-1, H_α_-8, H-10 and the isopropenyl unit were *α*-oriented. According to the coupling constants in the ^1^H-NMR, δ_H_ 2.39 (*J* = 13.9, 17.3 Hz) was assigned as H-2β because the dihedral angle of H-2β and H-1 should be close to 180, thus δ_H_ 2.46 (*J* = 5.1, 17.3 Hz) was assigned as H-2α [29]. The observed NOESY correlations between H-2β/H-7 further proved this. The relative configuration of 6-OH has been provisionally determined as α on the basis of alike chemical shift values of known compound **4**, for which the configuration had been determined by X-ray experiment [6]. 

The absolute configuration could be determined by the application of helicity of α-β unsaturated ketone as described previously [30]. The observed negative cotton effect at 248 nm (Appendix A) was consistent with the helicity rule, hence the configuration of **1** was deduced to be (–)-1*S*,6*R*,7*R*,9*R*,10*R*-6,9,15-trihydoxycadinan-4-en-3-one.

(+)-3,11,12-Trihydroxycalamenene (**2**) was prepared as a white amorphous powder and possessed a molecular formula of C_15_H_22_O_3_, which was determined by HRESIMS (*m/z* 273.1471, [M + Na]^+^ calcd. 273.1461). IR spectra revealed the presence of hydroxyl (3385 cm^−1^) and phenolic (1620 cm^−1^) groups. The ^1^H-NMR spectrum (Table 1) indicated two aromatic (δ_H_ 6.66, 7.12, s each) hydrogens, and two singlet (δ_H_ 1.14, 2.20, s each) and one doublet methyl (δ_H_ 1.24, d *J* = 7) groups.

The ^13^C-NMR and DEPT spectra (Table 1) exhibited fifteen carbon signals, including three methyls (δ_C_ 15.5, 21.6, 22.7), three methylenes (δ_C_ 23.0, 31.2, 67.7), four methines (δ_C_ 32.4, 44.0, 112.8, 133.1), and five quaternary carbons (δ_C_ 76.5, 120.7, 127.8, 133.1, 152.3). Two aromatic methine signals (δ_C_ 112.8, δ_H_ 6.66 s), (δ_C_ 133.1, δ_H_ 7.12 s) together with four aromatic carbons (δ_C_ 120.7, 127.8, 133.1, 152.3) suggesting a tetra-substituted aromatic spin system with a *para* configuration. Consideration of the unsaturation degrees and chemical shift characteristics, it was unambiguously proposed that compound **2** should be a bicyclic sesquiterpene with the calamenene skeleton [31,32,33]. 

The HMBC correlations of H-5 (δ_H_ 7.12) with C-15 (δc 15.5), and C-7 (δ_C_ 44.0), and of H-2 (δ_H_ 6.66) with C-10 (δ_C_ 32.4), and C-4 (δ_C_ 120.7) indicated that the two cycles were connected from C-7, and C-10, which allowed us to fully determine the aromatic moiety [31]. The phenolic hydroxyl group at C-3 was assigned by the HMBC correlations of H_3_-15 (δ_H_ 2.20) to C-3 (δ_C_ 152.3) and C-5 (δ_C_ 133.1). Meanwhile, the correlations of H_2_-12 (δ_H_ 3.57, 3.42, d J = 11.1) with C-7 (δ_C_ 44.0), C-11 (δ_C_ 76.5), and C-13 (δ_C_ 22.7), and of H_3_-13 (δ_H_ 1.14) with C-7 (δ_C_ 44.0), C-11(δ_C_ 76.5), and C-12 (δ_C_ 67.7) in the HMBC spectra corroborated the attachment of 1,2-isopropanediol group at position C-7. Furthermore, the HMBC correlations of H_3_-14 (δ_H_ 1.24, d *J* = 7.1) to C-1 (δ_C_ 143.7), C-9 (δ_C_ 31.2), and C-10 (δ_C_ 32.4), along with COSY cross peaks between H_3_-14(δ_H_ 1.24, d *J* = 7.1)/H-10 (δ_H_ 2.64, dq *J* = 14.2, 7.0)/H_2_-9 (δ_H_ 1.23, 1.93)/H_2_-8 (δ_H_ 1.81, 2.00)/H-7 (*δ*_H_ 3.09) determined the cyclohexane ring. Hence the planar structure of compound **2** with the calamenene core skeleton [31] was established (Figure 3).

Comparison of the chemical shifts data of compound **2** with that of known compound 3,12-dihydroxycalamenene [31] suggesting that two compounds have a close resemblance in their planar structure, except that C-11 (δ_C_ 76.5) in compound **2** was found to be an oxygenated quaternary carbon instead of a methine, as indicated by the HMBC correlations of H-7 (δ_H_ 3.09), H_2_-12 (δ_H_ 3.42, 3.57), and H_3_-13 (δ_H_ 1.14) with C-11.

The relative configuration of **2** was assigned by detailed analysis of its NOESY spectra and the ^1^H-NMR coupling constants. The observed NOESY correlations between H_3_-14 (δ_H_ 1.24) and H-7 (δ_H_ 3.09) indicated that H_3_-14 and H-7 were co-facial, while H_3_-14 and 1,2-isopropanediol group were *trans* to each other [34,35,36,37], unlike the *cis* conformations [31,32,33]. The above deductions were further proved by the NOESY cross peaks between H-10 (δ_H_ 2.64) and H_3_-13 (δ_H_ 1.14). The relative stereochemistry at the cyclohexene ring was also deduced from the ^1^H coupling constants [34]. The coupling constants between H-7 and H-8α and between H-10 and H-9β were 8.6 and 14.2 Hz, respectively, showing that these hydrogens have a *trans*-diaxial relationship, while the CH_3_-14 and the 1,2-isopropanediol group must both be equatorial and *trans*-oriented to each other [34]. OR experiment approved the compound to be dextrorotatory form (+12.1), very similar to the reported compound (+)-(1*S*,4*R*)-7-methoxycalamenene [34,35]. Hence the compound was named as (+)-(7*S*, 10*S*)-3,11,12-trihydroxycalamenene. The absolute configuration of C-11 remained to be determined. CD spectra of **2** (Appendix A) was also recorded to give the information for the further research, which displayed a negative cotton effect at 238 nm and 276 nm.

Compound 3 (–)-3,10,11,12-tetrahydroxycalamenene was isolated as white amorphous powder and its molecular formula C_15_H_22_O_4_ was established based on its HRESIMS data (*m/z* 289.1396, [M + Na]^+^ calcd. 289.1410). The IR spectra revealed the presence of hydroxyl (3379 cm^−1^) and phenolic (1623 cm^−1^) groups. The ^1^H-NMR spectrum (Table 1) indicated two aromatic (δ_H_ 6.68, 6.93, s each) hydrogen, and three methyl (δ_H_ 1.42, 1.57, 2.25, s each) groups.

The ^13^C-NMR and DEPT spectra (Table 1) exhibited fifteen carbon signals, including three methyls (δ_C_ 15.6, 22.0, 22.5), three methylenes (δ_C_ 21.0, 32.5, 69.1), three methines (δ_C_ 38.9, 108.1, 126.7), and six quaternary carbons (δ_C_ 72.3, 75.9, 122.0, 133.6, 140.6, 152.2). The similarity of the ^1^H and ^13^C-NMR spectroscopic data of **3** with those of **2** (Table 1) suggested that they were analogues. Detailed comparison of the chemical shifts showed that C-10 in **3** was an oxygenated quaternary carbon (δ_C_ 72.3) rather than a methine (δ_C_ 32.4, δ_H_ 2.64, qd, *J* = 14.2, 7.0) in **2**. This inference was further confirmed by the HMBC correlations of H_3_-14 (δ_H_ 1.57) and H-2 (δ_H_ 6.68) to C-10 (δ_C_ 72.3). The whole planar structure of **3** was determined (Figure 4) by referencing the compound **2,** and by surveying the 2D NMR spectral data.

The relative configuration of **3** was assigned by detailed analysis of its NOESY spectra, in which H_3_-14 (δ_H_ 1.57) was correlated with H-7 (δ_H_ 2.79, t, *J* = 2.7), illustrating the β-orientation of H-7, while the 1,2-isopropanediol group was *α*-oriented. The absence of correlations between H_3_-14, H_3_-13 and H_2_-12 also suggested that H_3_-14 and the 1,2-isopropanediol group were in *trans* conformation [34,35,36,37]. The ^1^H coupling constant between H_2_-8 and H-7 were 2.7 Hz in triplicate, indicating that H-7 and H-8α, H-8β are in *ee* and *ae* formation. According to 3D molecular model and coupling constant of H-7, the 1,2-isopropanediol group and 10-OH are in an axial position, which may be induced by the Van der Waals force between the 10-OH and 11-OH or 12-OH. The conformation change may also have resulted in the up fields shift of H_2_-12 from (*δ*_H_ 3.57, 3.42) in **2** to (δ_2H_ 2.85) in **3**, and the positive cotton effect at 263 nm in CD spectra (Appendix A) of **3**. OR experiment approved the compound to be a laevoisomer, thus the compound was named as (–)-(7*S*, 10*R*)-3,10,11,12-tetrahydroxycalamenene.

### 2.2. Antioxidant Activity of Isolated Compounds

The antioxidant bioassay of compounds (**1**–**8**) was performed on human whole blood neutrophils initially, in which **8** showed potential to inhibit the generation of reactive oxygen species (ROS). Further study on isolated human polymorphonuclear cells or neutrophils exhibited that **8** possessed strong antioxidant effects with an IC_50_ value = 3.68 ± 0.22 µM, comparing to the positive control Trolox with IC_50_ value = 77.23 ± 1.62 µM. 

### 2.3. Cytotoxic Activity of Isolated Compounds

Compounds **2**–**8** were also evaluated for their cytotoxicity against human tumor cells: HL-60 (acute leukemia), HepG-2 (liver hepatocellular carcinoma), HCT-16 (colorectal carcinoma), BGC-823 (gastric carcinoma cell) using MTT assay. Unfortunately, all the tested compounds showed the IC_50_ value at more than 100 µM. Since different biochemical, molecular and genetic mechanisms are involved in the development and progression of these cancers, the treatment also requires the specific agents. Ideally, the test compounds should be assayed for their cytotoxic potential on large number of cancer cell lines in the preliminary screening, with the aim of finding a lead compound to study the mechanism of action to treat such malicious tumors, which might be a limiting factor for any significant cytotoxicity that is lacking in our study [38].

## 3. Materials and Methods 

### 3.1. Plant Material

The stems of *Kadsura heteroclita* were collected in Shimen, Hunan, China, in September, 2014. The plant was identified by Prof. Wei Wang, and a voucher specimen (KH-shimen-201409) has been deposited in the TCM and Ethnomedicine Innovation & Development International Laboratory, Innovative Materia Medica Research Institute, School of Pharmacy, Hunan University of Chinese Medicine.

### 3.2. General and Solvents

Optical rotations were measured on a Perking-Elmer 341-MC digital polarimeter. UV spectra were recorded on a TU-1900 spectrophotometer (Shimadzu Europa GmbH, Duisburg, Germany). Experimental CD spectra were recorded on a JASCO J-815 Circular Dichroism (CD) Spectropolarimeter (Jasco, Mary’s Court Easton, MD, USA). A Hitachi 260-30 spectrometer was used for scanning IR spectroscopy. ID and 2D NMR spectra were performed on Bruker ARX-600 spectrometers ((Bruker Technology Co., Ltd., Karlsruhe, Germany) using TMS as the internal standard, with chemical shifts (δ) expressed in ppm and referenced to the solvent signals. HRESIMS were performed on a UPLC/xevo G2 Qtof spectrometer (Waters Corporation, Milford, MA, USA). Column chromatography (CC) was performed with silica gel (100–200 mesh and 200–300 mesh, Qingdao Marine Chemical, Inc., Qingdao, China), and Sephadex LH-20 (Pharmacia). Semipreparative HPLC was performed on an Agilent 1100 liquid chromatograph (Agilent Technologies, Santa Clara, CA, USA) with an Alltima C_18_ (5μ ODS 10 mm × 250 mm) column. Fractions were monitored by thin-layer chromatography (TLC) (Qingdao Marine Chemical Inc., Qingdao, China), and spots were visualized by heating the TLC silica gel plates sprayed with 10% H_2_SO_4_ in EtOH (10:90, *v/v*). Petroleum ether (PE), ethyl acetate (EtOAc), *n*-butanol (*n*-BuOH), dichloromethane (CH_2_Cl_2_) and methanol (MeOH) were purchased from Shanghai Titan Scientific Co., Ltd, Shanghai, China. MeOH (HPLC grade) and Water (HPLC grade) were purchased from Merck KGaA, 64271 Darmstadt, Germany.

### 3.3. Extraction and Isolation

The air-dried stems and roots of *K. heteroclita* (400 kg) were powdered and extracted with 95% aqueous ethanol (3 × 400 L, 3 days each) at room temperature and concentrated under reduced pressure to afford a crude extraction (8 kg).

1 kg of crude extraction was suspended in H_2_O (10 L) and was partitioned between water and *n*-BuOH (10 L) to afford the *n*-BuOH fraction, repeated for 3 times, the *n*-BuOH layer was combined and evaporated to yield a residue of 251.5 g. Then 240.0 g of *n*-BuOH fraction was chromatographed on a silica gel column (3.0 kg, 100–200 mesh) using dichloromethane/methanol as elution solvents (from 99.8:0.2 to 0:100 gradient system), to afford fraction A–J.

Fraction C (35.1 g) was chromatographed using silica gel CC (0.9 kg, 200–300 mesh), eluted with a DCM/MeOH gradient system (99.8:0.2 to 0:100) to obtain ten fractions (Fr. C-1 to Fr. C-10). Fr. C-5 (3.1 g) was loaded to silica gel (0.1 kg, 200–300 mesh) column and eluted with petroleum ether/ethyl acetate (90:10 to 20:80) to produce eight fractions (Fr. C-5-a to Fr. C-5-h). Fr. C-5-b (227.0 mg) was isolated by silica gel column (9.0 g, 200–300 mesh), by eluting with PE/EtOAc (40:60) to afford compound **5** (12.0 mg). Fr. C-10 (2.0 g) was subjected to silica gel CC (0.15 kg, 200–300 mesh) eluted with PE/EtOAc (80:20 to 0:100) to give twelve fractions (Fr. C-10-1 to Fr. C-10-12). Fr. C-10-7 was purified by semipreparative HPLC, with a solvent of MeOH/H_2_O (3 ml/min, 75:25), to afford **8** (4.3 mg).

Fraction D (20.6 g) was separated over silica gel CC (0.6 kg, 200–300 mesh) with a gradient elution of DCM/MeOH (99.5:0.5 to 0:100) to give ten fractions (Fr. D-1 to Fr. D-10). Fr. D-3 (1.5 g) was repeated eluting on silica gel CC (0.2 kg, 200–300 mesh) and produced six subfractions (D-3-a to D-3-f), D-3-d (350.0 mg) was purified by silica gel CC (50.0 g, 200–300 mesh) eluting with PE/EtOAc (90:10 to 0:100) to yield two crystals **4** (6.5 mg) and **6** (120.5 mg). Fraction D-5 (1.2 g) was chromatographed by silica gel CC (0.1 kg, 200–300 mesh) eluted with PE/EtOAc (90:10 to 0:100) to give six fractions (Fr. D-5-a to Fr. D-5-f). Fr. D-5-a (214.7 mg) was applied to another silica gel CC (20.0 g, 200–300 mesh) eluted with PE/EtOAc (80:20) to give six fractions (Fr. D-5-a-1 to Fr. D-5-a-6). Fr. D-5-a-5 (42.0 mg) was purified by semipreparative HPLC (2 mL/min, MeOH/H_2_O, 75:25) to yield **3** (1.8 mg) and **2** (7.6 mg) respectively. The Fr. D-6 (1.0 g) was repeated on silica gel (0.1 kg, 200–300 mesh) column eluting with PE/EtOAc (50:50 to 0:100) to yield eight fractions (Fr. D-6-a to Fr. D-6-h). Fr. C-6-f (201.3 mg) was separated by sephadex LH 20 (10.0 g) column with the washing solvent methanol/chloroform (1:1), to produce four fractions (Fr. C-6-f-1 to C-6-f-4). Fr. C-6-f-2 (21.7 mg) was purified by semi prepare HPLC (MeOH/H_2_O, 75: 25) to obtain compound **1** (2.1 mg). Fr. D-7 (0.5 g) was subjected to silica gel CC (50.0 g, 200–300 mesh) eluted with PE/EtOAc (80:20 to 0:100) to provide eight fractions (Fr. D-7-a to Fr. D-7-h). Fr. D-7-g (54.8 mg) was purified by semipreparative HPLC, with a solvent system of MeOH/H_2_O (3 mL/min, 70:30), to afford compounds **7** (10.2 mg). 

### 3.4. Spectroscopic Data of New Compounds

*6α,9α,15-Trihydroxycadinan-4-en-3-one* (**1**): white solid (MeOH); ${\left[\alpha \right]}_{D}^{20}$ −20.0 (*c* 0.25, MeOH); UV(MeOH) *λ*_max_ (log *ε*) 230 (3.10) nm; CD (*c* 0.50, MeOH) *λ*_max_ (∆*ε*) 248 (−4.42), 270 (+3.66), 332 (+4.99) nm; IR *ν*_max_ 3410, 2930, 1713, 1668, 1606, 1283, 1255, 1098 cm^−1^; ^1^H and ^13^C-NMR data, see Table 1; positive HRESIMS *m*/*z* 289.1419 [M + Na] ^+^ (calcd for C_15_H_22_O_4_Na, 289.1416).

*(+)-3,11,12-trihydroxycalamenene* (**2**): white, amorphous solid; ${\left[\alpha \right]}_{D}^{20}$ +12.1 (*c* 1.9, MeOH); UV(MeOH) *λ*_max_ (log *ε*) 283 (3.41) nm; CD (*c* 0.33 MeOH) *λ*_max_ (∆*ε*) 238 (−5.49) nm, 276 (−8.68); IR *ν*_max_ 3385, 2960, 2932, 2873, 1620, 1505, 1457, 1381, 1260, 1187, 1114, 1039 cm^−1^; ^1^H and ^13^C-NMR data, see Table 1; positive HRESIMS *m*/*z* 273.1471 [M + Na] ^+^ (calcd for C_15_H_22_O_3_Na, 273.1461).

*(–)-3,10,11,12-tetrahydroxy-calamenene* (**3**): white, amorphous solid; ${\left[\alpha \right]}_{D}^{20}$ −2.2 (*c* 0.45, MeOH); UV(MeOH) *λ*_max_ (log *ε*) 220 (3.90) nm, 280 (3.45) nm; CD (*c* 0.12, MeOH) *λ*_max_ (∆*ε*) 215 (+11.85), 237 (−2.42), 263 (+6.16) nm; IR *ν*_max_ 3379, 2932, 1623, 1463, 1421, 1376, 1196, 1044 cm^−1^; ^1^H and ^13^C-NMR data, see Table 1; positive HRESIMS *m*/*z* 289.1396 [M + Na] ^+^ (calcd for C_15_H_22_O_4_Na, 289.1416).

### 3.5. Antioxidant Assay 

Chemiluminescence assay was utilized to measure the antioxidant activity. It is a sensitive method to measure the inhibition of ROS production [39]. Phorbol 12-myristate 13-acetate (PMA) is used as an inducer for the generation of various ROS by macrophages and neutrophils, and PMA is used as an activator of protein kinase C and activator of NADPH oxidase in our study [11].

Different concentrations of samples were incubated with either diluted whole blood 40 µL (1:25 dilution in sterile PBS, pH 7.4) or PMN 40 µL (Polymorphonuclear cells, 1 × 10^6^/mL) suspended in HBSS^++^. The cells were stimulated with 40 µL of opsonized zymosan followed by luminol as enhancer and then HBSS^++^ was added to adjust the final volume to 200 µL. The final concentrations of the samples in the mixture were 20, 10, 5, 2.5 and 0.625 µM. Microplates were then incubated at 22 ℃ for 30 min. The results were measured by Enspire Multimode Plate Reader (PerkinElmer, Inc., Winter Street Waltham, MA, USA), as counts per second (CPS). Inhibition percentage (%) calculated as:

Inhibition percentage (%) = 100 − (CPS_test_/CPS_control_) × 100

### 3.6. Cytotoxicity Assays

The cytotoxicity assay was performed using an MTT assay [40], the experiment method referenced to the reported literature [11] and the following human tumor cell lines were used: HL-60 (acute leukemia), HepG-2 (liver hepatocellular carcinoma), HCT-16 (colorectal carcinoma), BGC-823 (gastric carcinoma cell). Each tumor cell line was exposed to test compounds in triplicate for 48 hr., with Taxol used as a positive control substance. 

## Figures and Tables

**Figure 1 molecules-24-01664-f001:**
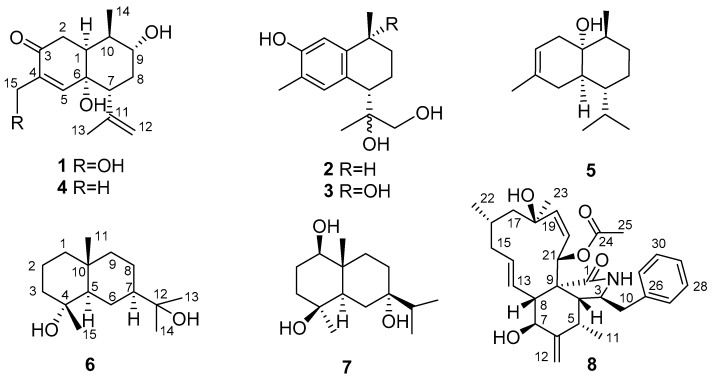
Structures of compounds **1**–**8**.

**Figure 2 molecules-24-01664-f002:**
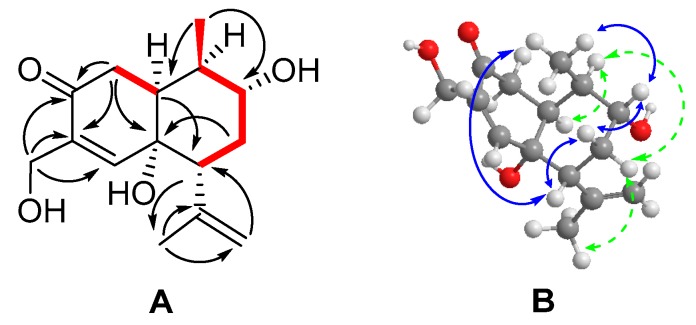
^1^H−^1^H COSY, HMBC and NOESY correlations of compound **1**. **A**: Key ^1^H−^1^H COSY (bold ─), and HMBC (→) correlations. **B**: Key NOESY (
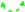
) correlations.

**Figure 3 molecules-24-01664-f003:**
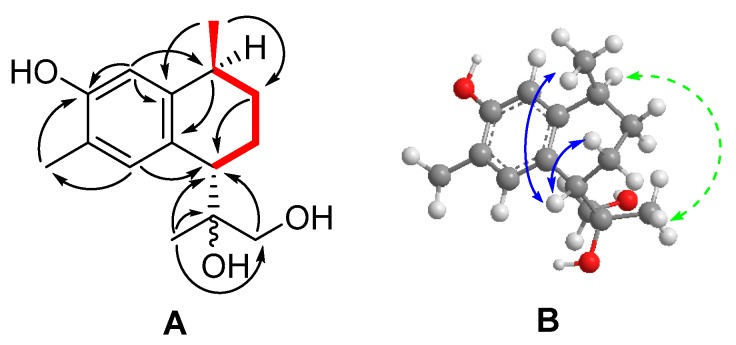
^1^H−^1^H COSY, HMBC and NOESY correlations of compound **2**. A: Key ^1^H−^1^H COSY (bold ─), and HMBC (→) correlations. B: Key NOESY (
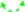
) correlations.

**Figure 4 molecules-24-01664-f004:**
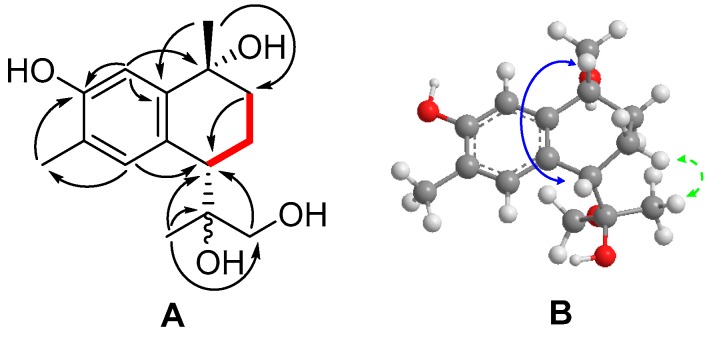
^1^H−^1^H COSY, HMBC and NOESY correlations of compound **3**. **A**: Key ^1^H−^1^H COSY (bold ─), and HMBC (→) correlations. **B**: Key NOESY (
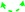
) correlations.

**Table 1 molecules-24-01664-t001:** ^1^H and ^13^C-NMR spectroscopic data for compound **1**–**3** (*δ* in ppm, *J* in Hz).

Num.	1 ^a^		2 ^b^		3 ^b^	
	δ_H_	δ_C_	δ_H_	δ_C_	δ_H_	δ_C_
1	2.21 (dt, 13.9, 4.9)	50.2		143.7		140.6
2α	2.46 (dd, 5.1, 17.3)	37.0	6.66 (s)	112.8	6.68 (s)	108.1
2β	2.39 (dd, 13.9, 17.3)					
3		200.5		152.3		152.2
4		137.0		120.7		122.0
5	6.85 (s)	155.2	7.12 (s)	133.1	6.93 (s)	126.7
6		73.3		127.8		133.6
7β	2.54 (dd, 13.4 3.3)	48.6	3.09 (dd, 8.6, 5.6)	44.0	2.79 (t, 2.7)	38.9
8α	2.00 (m)	36.8	2.00 (m)	23.0	2.23 (dt,12.4 2.9)	21.0
8β	1.65 (m)		1.81 (m)		1.35 (m)	
9α		71.9	1.23 (m)	31.2	1.39 (dd, 11.8, 2.2)	32.5
9β	3. 47 (td, 10.9, 4.4)		1.93 (m)		2.00 (m)	
10α	2.14 (m)	37.3	2.64 (dq, 14.2 7.0)	32.4		72.3
11		146.6		76.5		75.9
12a	4.77 (s)	114.6	3.57 (d 11.1)	67.7	2.85 (s)	69.1
12b	4.90 (s)		3.42 (d 11.1)		2.85 (s)	
13	1.83 (s)	23.0	1.14 (s)	22.7	1.42 (s)	22.5
14	1.00 (d 6.9)	15.0	1.24 (d 7.1)	21.6	1.57 (s)	22.0
15	4.16 (d 1.14)	59.5	2.20 (s)	15.5	2.25 (s)	15.6

^a^ Recorded at 600 MHz in CD_3_OD. ^b^ Recorded at 600 MHz in CDCl_3_.

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
