# Peer review of "New Cadinane Sesquiterpenes from the Stems of Kadsura heteroclita"

_molecules, 2019, doi:10.3390/molecules24091664_

Round 1

Reviewer 1 Report

I think that the paper under review deserves publication in Molecules. The topic is of interest for the journal readers and the described findings are new and scientifically sound. In spite of this fact, I deem that the paper contains a number of errors/inaccuracies that reduce the quality of the work and should be amended before publication. I would like suggesting the following integrations and modifications:-        Abstract. Line 9: ‘Affiliation 1’ should be removed-        Abstract. The authors indicated the evaluation of the anti-inflammatory activity despite only antioxidant and cytotoxicity assays are described in the experimental section. Although the anti-inflammatory activity is often related to the antioxidant activity, the two assays do not measure the same thing.-        Line 109. The term ‘hence’ is written with a different font.-        Line 122. The term ‘trans’ should be written with italic style.-        Line 122-126. It is not clear how was determined the absolute configuration of compound 2. Please elaborate.-        Compounds 1-3 are solid. Please measure their melting points.

Throughout the paper. I think that the paper needs of a general brush-up on English language.

Author Response

Response to Reviewer 1 Comments

No.: molecules-471070

Title: New Cadinane Sesquiterpenes from the Stems of Kadsura heteroclita

Thank you for your kind comments concerning our manuscript. Those comments are all valuable and very helpful to improve our manuscript. We have studied comments carefully and have made the corrections point by point accordingly. The main corrections in the revision and the responds to your comments are as follows:

Point 1: Abstract. Line 9: ‘Affiliation 1’ should be removed.

Response 1: It is a typo. We removed ‘Affiliation 1’.

Point 2: Abstract. The authors indicated the evaluation of the anti-inflammatory activity despite only antioxidant and cytotoxicity assays are described in the experimental section. Although the anti-inflammatory activity is often related to the antioxidant activity, the two assays do not measure the same thing.

Response 2: In abstract, anti-inflammatory term is changed to antioxidant.

Point 3: Line 109. The term ‘hence’ is written with a different font.

Response 3: We reset the front of “Hence” (Page 4 Line 121)

Point 4: Line 122. The term ‘trans’ should be written with italic style.

Response 4: We have adjusted the “trans” to italic style (Page 4 Line 133)

Point 5: Line 122-126. It is not clear how was determined the absolute configuration of compound 2. Please elaborate.

Response 5: Since compound 2 has a known calamenene skeleton, we only need to discuss the relative configuration of moieties.

Point 6: Compounds 1-3 are solid. Please measure their melting points.

Response 6: The melting points of compounds 1-3 are not measured, because these compounds are not crystal.

Point 7: Throughout the paper. I think that the paper needs of a general brush-up on English language.

Response 7: We have invited some native English speak researchers in the area to polish the English writing.

Reviewer 2 Report

The manuscript by Cao et al., is very well written and presented as well. I recommend publication in molecules. However, minor points should be considered:

1- Althought the similarity test revealed high percentage (42%), from my point of view only section 3.5 should be paraphrased.

2- P2 L 59 " carbonyl the remaining", revise please.

3- In figure 1. compounds 2 and 3 an be combined.

4- P6 L178 use italic "n" and subscripts.

Author Response

Response to Reviewer 2 Comments

No.: molecules-471070

Title: New Cadinane Sesquiterpenes from the Stems of Kadsura heteroclita

Thank you for your kind comments concerning our manuscript. Those comments are all valuable and very helpful to improve our manuscript. We have studied comments carefully and have made the corrections point by point accordingly. The main corrections in the revision and the responds to your comments are as follows:

Point 1: Althought the similarity test revealed high percentage (42%), from my point of view only section 3.5 should be paraphrased.

Response 1: Heading 3.5 is paraphrased. (Page 7 Line 246-258)

Point 2: P2 L 59 " carbonyl the remaining", revise please.

Response 2: This sentence has been modified to " carbonyl, the remaining"(Page 2 Line 66)

Point 3: In figure 1. compounds 2 and 3 can be combined.

Response 3: We’d like to accept the suggestion. (Page 2 Line 53)

Point 4: P6 L178 use italic "n" and subscripts.

Response 4: We have adjusted the “n” to italic style (Page 6 Line 195).

Reviewer 3 Report

The paper entitled New Cadinane Sesquiterpenes from the Stems of Kadsura heteroclite on phytochemical investigation of Kadsura heteroclite in my opinion cannot be accepted for publication on Molecules

In particular, new sesquiterpenes reported (1-3) don’t show many structural differences between other know compounds (for example 6α,9α-dihydroxycadinan- 4-en-3-one) as authors reported.

 Great effort was made to purify little amounts of all compounds. Authors extracted 400 kg of air- dried stems and used a big amount of solvent and a very long purification process but  NMR spectra show that 1-3 and 5 are not pure.

About stereochemistry in my opinion some comments on NoE effects are not fully clear. How the authors assigned chemical shifts at H-8 in alfa and beta position in 1?

For 2 the sentence is not clear “H3-14 (δH 1.57) correlated to H-2 (δH 6.68) and H-7 (δH 2.79), and H-5 (δH 6.93) correlated to H-7 (δH 2.79), H3-15 (δH 2.25) illustrating that H-7 and H3-14 are in β-orientation, while the 1,2-isopropanediol group and 10-OH were α-oriented”. H-2 and H-5 are planar so their correlations between H3-14 and H-7 are not significant.  The same consideration applies to 3.

The authors also recorded circular dichroism (see Spectroscopic data of new compounds) but no information was given on this respect. Is it possible compare CD data with those of related know compounds?

I suggest submitting paper to journal with lower IF.

Author Response

Response to Reviewer 3 Comments

No.: molecules-471070

Title: New Cadinane Sesquiterpenes from the Stems of Kadsura heteroclita

Thank you for your kind comments concerning our manuscript. Those comments are all valuable and very helpful to improve our manuscript. We have studied comments carefully and have made the corrections point by point accordingly. The main corrections in the revision and the responds to your comments are as follows:

Point 1: new sesquiterpenes reported (1-3) don’t show many structural differences between other know compounds (for example 6α,9α-dihydroxycadinan- 4-en-3-one) as authors reported.

Response 1: So far, there are seldom sesquiterpenes have been reported from the genus of Kadsura, especially with the Cadinene skeleton. And Cadinane sesquiterpenes with 1,2-isopropanediol group are novel both in the structure and in stereochemistry determination work (please see reference 27 and 28).

Point 2: Great effort was made to purify little amounts of all compounds. Authors extracted 400 kg of air- dried stems and used a big amount of solvent and a very long purification process but NMR spectra show that 1-3 and 5 are not pure.

Response 2: We really did hard work for the isolation work and did our best to purify these compounds.

Point 3: About stereochemistry in my opinion some comments on NoE effects are not fully clear. How the authors assigned chemical shifts at H-8 in alfa and beta position in 1?

Response 3: We agree with the referee’s opinion. It is not necessary to distinguish alfa and beta position of H2-8. So we change alfa and beta of H-8 and -9 to a and b respectively (see Table 1).

Point 4: For 2 the sentence is not clear “H3-14 (δH 1.57) correlated to H-2 (δH 6.68) and H-7 (δH 2.79), and H-5 (δH 6.93) correlated to H-7 (δH 2.79), H3-15 (δH 2.25) illustrating that H-7 and H3-14 are in β-orientation, while the 1,2-isopropanediol group and 10-OH were α-oriented”. H-2 and H-5 are planar so their correlations between H3-14 and H-7 are not significant. The same consideration applies to 3.

Response 4: We totally agree with the viewpoints of the referee. Therefore “H3-14 (δH 1.57) correlated to H-2 (δH 6.68) and H-7 (δH 2.79), and H-5 (δH 6.93) correlated to H-7 (δH 2.79), H3-15 (δH 2.25) illustrating that H-7 and H3-14 are in β-orientation, while the 1,2-isopropanediol group and 10-OH were α-oriented” was corrected as “H3-14 (δH 1.57) was correlated with H-7 (δH 2.79), illustrating the β-orientation of H-7, while the 1,2-isopropanediol group was α-oriented.” accordingly (see the revision page 4 Line 131-132).

Point 5: The authors also recorded circular dichroism (see Spectroscopic data of new compounds) but no information was given on this respect. Is it possible compare CD data with those of related know compounds?

Response 5: We just provided and recorded the CD spectra data of new compounds. Hope it might be the references for the further stereochemistry research if necessary.

Point 6: I suggest submitting paper to journal with lower IF.

Response 6: The authors thanks the referee’s very good questions. We have double-checked and finished our major revision according to all the requirements.

Reviewer 4 Report

Minor revision. Pretty nice, indeed. As a consequence, my truly congrats go to all authors.

If possible, please, kindly improve a bit the English language. Also, some typos throughout the text of Your valuable manuscript (MS) need to be properly addressed prior to its publishing in Molecules.

In addition to this, You may kindly consider citing of the following references within Your MS, aiming to briefly discuss lack of any significant cytotoxicity:

Curr Top Med Chem. 2013;13(21):2745-66.

Nat Prod Res. 2014;28(24):2237-44.

Last but not least, very best of (research) luck ahead!

Author Response

Response to Reviewer 4 Comments

No.: molecules-471070

Title: New Cadinane Sesquiterpenes from the Stems of Kadsura heteroclita

Thank you for your kind comments concerning our manuscript. Those comments are all valuable and very helpful to improve our manuscript. We have studied comments carefully and have made the corrections point by point accordingly. The main corrections in the revision and the responds to your comments are as follows:

Response 1: We found out some typos in the manuscript and subsequently corrected. And also invite some native English speak researchers to revise the writing.

Point 2: In addition to this, you may kindly consider citing of the following references within Your MS, aiming to briefly discuss lack of any significant cytotoxicity:

- Curr Top Med Chem. 2013;13(21):2745-66.

- Nat Prod Res. 2014;28(24):2237-44.

Response 2: Thank you very much. We have discussed briefly the reason of lacking any cytotoxicity in our study, with proper citation (Page 9, Line 368-370), as per your valuable suggestions.

Round 2

Reviewer 3 Report

The second version of manuscript doesn’t present many improvements, for example CD data interpretation is lacking.

My doubts remain, and I suggest again to send it to journal with lower IF

Author Response

Response 1: We have done major revise for the manuscript, for example:

1) the stereochemistry of 1 has been solved by referred to the CD spectra and 1H NMR coupling constant (P3, L96-99, L90-92).

2) The order of the new compounds has been changed [(+)-3,11,12- trihydroxycalamenene (2), before revision is compound 3; (–)-3,10,11,12-tetrahydroxy-calamenene (3) before revision is compound 2.] and also modified the supporting information.

3) the stereochemistry of compound 2 and 3 were resolved by analysis of 1H NMR coupling constant and NOESY spectra, and by refereeing to the literatures. CD spectra were discussed in this reversion (P4, L139-147; P5, L171-177).

The authors thank the referee’s very good questions. We have double-checked and finished our major revision according to all the requirements. Hope it will be arrived at the acceptation of dear reviewer.